# Effect of Heat Treatment on Protein Self-Digestion in Ruminants’ Milk

**DOI:** 10.3390/foods12183511

**Published:** 2023-09-21

**Authors:** Juliana A. S. Leite, Carlos A. Montoya, Evelyne Maes, Charles Hefer, Raul A. P. A. Cruz, Nicole C. Roy, Warren C. McNabb

**Affiliations:** 1Riddet Institute, Te Ohu Rangahau Kai Facility, Massey University, Palmerston North 4474, New Zealand; ju_leite_@hotmail.com (J.A.S.L.); carlos.montoya@agresearch.co.nz (C.A.M.); evelyne.maes@agresearch.co.nz (E.M.); nicole.roy@otago.ac.nz (N.C.R.); 2Smart Foods & Bioproducts Innovation Centre of Excellence, AgResearch Limited, Palmerston North 4474, New Zealand; 3Proteins & Metabolites Team, AgResearch Lincoln Research Centre, Lincoln 7608, New Zealand; 4Data Science South Team, AgResearch Lincoln Research Centre, Lincoln 7608, New Zealand; charles.hefer@agresearch.co.nz; 5School of Food & Advanced Technology, Massey University, Palmerston North 4442, New Zealand; raulacedo@msn.com; 6High-Value Nutrition National Science Challenge, Auckland 1010, New Zealand; 7Department of Nutrition, University of Otago, Dunedin 9016, New Zealand

**Keywords:** ruminant milk, proteases, protein hydrolysis, self-digestion, heat treatment, peptides, bovine milk, ovine milk, caprine milk, free amino groups

## Abstract

This study investigated whether heat treatments (raw, 63 °C for 30 min, and 85 °C for 5 min) affect protein hydrolysis by endogenous enzymes in the milk of ruminants (bovine, ovine, and caprine) using a self-digestion model. Self-digestion consisted of the incubation for six hours at 37 °C of the ruminants’ milk. Free amino group concentration was measured by the *o*-phthaldialdehyde method, and peptide sequences were identified by chromatography-mass spectrometry. Results showed that heat treatments prior to self-digestion decreased the free NH_2_ by 59% in bovine milk heated at 85 °C/5 min, and by 44 and 53% in caprine milk heated at 63 °C/30 min and 85 °C/5 min, respectively. However, after self-digestion, only new free amino groups were observed for the raw and heated at 63 °C/30 min milk. β-Casein was the most cleaved protein in the raw and heated at 63 °C/30 min bovine milk. A similar trend was observed in raw ovine and caprine milk. Self-digestion increased 6.8-fold the potential antithrombin peptides in the bovine milk heated at 63 °C/30 min. Enhancing bioactive peptide abundance through self-digestion has potential applications in the industry for functional products. Overall, heat treatments affected the free amino groups according to the species and heat treatment applied, which was reflected in the varying degrees of cleaved peptide bonds and peptides released during self-digestion.

## 1. Introduction

Milk is an important source of essential nutrients and contains many proteins and bioactive peptides that can play an important role in the consumer’s health [1]. The protein composition and casein micelle structures in milk differ across species (bovine, ovine, and caprine), which impacts gastric curd formation and the degradation of individual proteins [2]. As shown in our previous works [3,4], the concentration of endogenous milk proteases also differs between species, and this difference might affect the hydrolysis of milk proteins during gastrointestinal digestion. For instance, raw ovine milk had 11-fold higher plasmin activity than raw bovine milk [3].

Several studies have shown that heat treatments applied during industrial processing to ensure the safety of dairy products can impact milk protein structures. These treatments induce protein degradation and aggregation [1,5,6,7] that may hinder or enhance proteolysis by endogenous milk proteases during gastrointestinal digestion.

Mild treatments applied to milk such as 63 °C for 30 min can increase protease activities due to the activation of zymogens [3,6], while more intense heat treatments such as 85 °C for 5 min can reduce their activities due to the inactivation of active enzymes through protein aggregation or denaturation [3,8,9]. Therefore, proteases are heat sensitive.

Although the effect of different heat treatments on protease activities of bovine milk has been well-studied [3,6,8,10,11], there is still a lack of information on the role of endogenous milk enzymes on self-digestion, mainly in ovine and caprine milk. Most studies have investigated the digestion of milk proteins using static and semi-dynamic in vitro gastrointestinal models [12,13]. These models normally use enzymes (mainly pepsin and pancreatin) that mask the role of endogenous milk proteases on the hydrolysis of milk proteins.

Understanding the role of endogenous milk enzymes under a self-digestion model, which simulates some physiological conditions of the human gastrointestinal tract lumen such as body temperature and digestion time can provide new insights to develop new dairy products with potential enhanced nutritional value (e.g., faster rate of digestion and absorption) and functionality (e.g., specific bioactive peptides). Our team has demonstrated that thermal treatments applied to milk could affect the realising of peptides with distinct functionalities [3]. However, a gap in our understanding remains, specifically concerning the impact of heat treatments on the abundance of bioactive peptides and whether changes in abundance occur during the digestion process due to the influence of endogenous milk enzymes, which can only be accurately measured without the interference of digestive enzymes.

The hypothesis of the study is that different heat treatments might alter the proteolytic system of ruminant milk, affecting the proteolysis of milk proteins to release specific bioactive peptides during self-digestion. This study investigated the effect of different treatments (raw, 63 °C for 30 min, and 85 °C for 5 min) on the self-digestion (37 °C for 6 h) of proteins in ruminant milk (bovine, ovine, and caprine). Free amino groups (NH_2_) were measured to determine the peptide bonds prior and post the self-digestion of raw and heat-treated milk. The peptide sequences on the ruminant milk samples prior and post self-digestion were characterised using high-resolution liquid chromatography tandem mass spectrometry (LC-MS/MS). Based on these peptide identifications, predictions were made demonstrating which milk proteins were hydrolysed by endogenous proteases, and which peptides could have been generated after self-digestion.

## 2. Materials and Methods

### 2.1. Milk Sampling

Fresh raw bovine (*Bos taurus*), ovine (*Ovis aries*), and caprine (*Capra hircus*) milk were collected during early lactation to minimise the variations in enzyme activities associated with lactation stage [3]. Raw bovine milk was provided by Dairy 4 Farm at Massey University in Palmerston North, New Zealand, while ovine and caprine milk were obtained from local dairy farms in Palmerston North and Bulls, New Zealand, respectively. The raw ruminant milk samples were used for both pasteurisation and self-digestion. Milk pasteurisation was carried out in the morning and self-digestion in the afternoon of the collection day. For each species, three batches of milk from different animals were collected, heat treated, and self-digested over three separate days.

### 2.2. Heat Treatments and Self-Digestion

Raw ruminant milk samples were first skimmed at 2455× *g* for 30 min at 4 °C to remove the fat. Subsequently, skim milk samples were pasteurised using two different conditions: 63 °C for 30 min or 85 °C for 5 min in a thermostatically controlled water-bath with shaking. These thermal conditions were chosen to represent the conventional pasteurisations using low-temperature, long-time (LTLT, 63 °C for 30 min) and high-temperature, short-time (HTST, 85 °C for 5 min). Moreover, the selection of these thermal conditions was based on the expectation that LTLT pasteurisation would result in less damage to the proteolytic system compared to HTST pasteurisation due to its milder nature.

For more reliable temperature monitoring, a thermometer (MS6514, Mastech, Dongguan, China) was inserted at the centre of a control bottle containing milk. Once the target temperature was reached, the incubation time began, and the desired temperature remained constant throughout the holding period. After achieving the specific time–temperature condition, the bottles with milk samples were rapidly removed from the water-bath and immersed in an ice-water bath until the temperature achieved approximately 10 °C [3]. Skim raw and pasteurised samples were aliquoted and frozen at −80 °C until the determination of the free NH_2_ group concentration and peptide profile.

To assess the proteolytic ability of endogenous proteases, skimmed raw and pasteurised ruminant milk were self-digested for 6 h at 37 °C in a water-bath with shaking, without pH adjustments or the addition of gastrointestinal enzymes (i.e., pepsin and pancreatin) to understand the action of endogenous milk proteases on the intact milk. The temperature of 37 °C was chosen to simulate the body temperature and 6 h of incubation as the transit time between the stomach and the small intestine [14,15]. Aliquots were collected after 6 h of incubation and immediately frozen at −80 °C until the determination of the free NH_2_ concentration and peptide profile.

### 2.3. Degree of Hydrolysis by Determination of Free Amino Groups

The degree of protein hydrolysis before and after 6 h of self-digestion was determined by measuring the concentration of the free NH_2_ group in the skimmed raw and pasteurised ruminant milk using the *o*-phthaldialdehyde (OPA) assay, as described elsewhere [16]. Briefly, the OPA assay was carried out by adding 100 µL of the sample to 1 mL of OPA solution (1.25 mL of 20% sodium dodecyl sulphate; 40 mg of OPA dissolved in 1 mL methanol; 100 µL of 2-mercaptoethanol, adjusted to a final volume of 50 mL with 0.1 M sodium tetraborate) in a cuvette. The absorbance was measured after exactly 2 min at 340 nm in a UV/Visible spectrophotometer (Genesys, Thermo Fisher Scientific, San Jose, CA, USA). A calibration curve was prepared using glycine standard solutions (0 to 1 mM). The concentration of free NH_2_ in the skimmed raw and heated ruminant milk samples at time 0 h was used as a control.

### 2.4. Peptide Analysis

#### 2.4.1. Peptide Extraction

Extraction of the endogenous peptides from the milk samples was performed as described in our previous study [3] with minor modifications. Briefly, 200 µL aliquots of all skimmed raw and heat-treated ruminant milk samples prior and post self-digestion were centrifuged at 14,000× *g* for 30 min at 4 °C. The clear supernatant was transferred into a new Eppendorf tube and diluted twice with 5% acetonitrile. Next, the samples were ultra-filtered using 10 kDa NanoSep centrifugal ultrafilters (Pall, Ann Arbor, MI, USA) at 14,000× *g* for 25 min at 4 °C. The ultrafiltrate was dried in a vacuum centrifuge and resuspended in 100 µL of 0.1% formic acid followed by protein quantitation using Nanodrop One (Thermo Fisher Scientific, San Jose, CA, USA). Peptide concentrations were normalised to 2 µg/ µl prior to the mass spectrometric analysis.

#### 2.4.2. Peptide Characterisation via LC-MS/MS

All samples were analysed with a nanoflow Ultimate 3000 RSLC (Thermo Scientific, San Jose, CA, USA) coupled to an Impact II mass spectrometer with a CaptiveSpray source equipped with a nanoBooster device (Bruker Daltonik, Bremen, Germany) operated at 1800 V. For each sample, 1 µL was loaded on a C18 PepMap100 Nano-Trap column (300 µm ID × 5 mm, 5 micron 100 Å) at a flow rate of 3000 nL/min. The trap column was then switched in line with the analytical column ProntoSIL C18AQ (100 µm ID × 150 mm 3-micron 200 Å) (nanoLCMS Solutions, Gold River, CA, USA). The reverse-phase elution gradient was from 2% to 20% to 45% solvent B over 60 min, a total of 88 min at a flow rate of 600 nL/min. Solvent A was LCMS-grade water with 0.1% formic acid; solvent B was LCMS-grade acetonitrile with 0.1% formic acid.

To quantitatively profile the peptides, the analysis of the samples was performed in positive ion MS mode, with a mass range between 150 and 2200 *m/z* and a sampling rate of 2 Hz. All samples from each species (including biological repeats) were randomised and analysed in duplicate as two separate batches.

To link the peptide abundance levels with identifications, a pool from each treatment group was made by combining an equal amount of the samples, and LC-MS/MS runs of these pools were performed using data-dependent acquisition with the following settings: the same LC parameters as described before, a full scan MS spectrum with a mass range of 50–2500 *m/z* was followed by a maximum of 10 collision-induced dissociation tandem mass spectra (150–2200 *m/z*) at a sampling rate of 2 Hz for MS scans and 1–20 Hz for MS/MS (depending on precursor intensity). Precursors with charges 1+ to 8+ were preferred for further fragmentation, and a dynamic exclusion of 60 seconds was set.

#### 2.4.3. Peptide Identification

The PEAKS X+ Studio data analysis software package (Bioinformatics Solutions Inc., Waterloo, ON, Canada) was used to analyse the LC-MS/MS data. The raw data were refined by a built-in algorithm that allows for the association of chimeric spectra. The peptides were identified with the following parameters: a precursor mass error tolerance of 10 ppm and fragment mass error tolerance of 0.05 Da were allowed; peptides with a length starting at four amino acids long were included; the Uniprot *Ovis aries* database (v2019.08, 27,855 sequences), Uniprot *Bos taurus* database (v2019.08, 46,707 sequences), and Uniprot *Capra hircus* database (v2019.08, 35,307 sequences) were used; no enzyme was specified as a digestive enzyme. Oxidation, phosphorylation, and deamidation were chosen as variable modifications in Peaks DB (database), and unexpected modifications were accounted for in the Peaks Post-Translational Modifications search module. A maximum of three post-translational modifications per peptide was permitted. The false discovery rate (FDR) estimation was made based on decoy-fusion. A FDR of less than 5% for confident peptide identification was set. The peptides identified were compiled for the heat treatment (raw, 63 °C/30 min, and 85 °C/5 min) and self-digestion time (0 and 6 h) combination within species. When peptides with the same amino acid sequences and retention time (i.e., redundant peptides) were identified multiple times in the same sample, they were considered as only one peptide. Peptide abundance refers to the area under the curve (AUC) of the eluted peak.

#### 2.4.4. Protein of Origin

The identified non-redundant peptides were grouped by protein of origin for each heat treatment and self-digestion time combination within species and their AUC were summed up by each protein. Not all of the peptides had their protein of origin identified because this study was designed to target endogenous peptides and not all possible peptides from a protein. Only the proteins with a high abundance (AUC) of non-redundant peptides in the digested skimmed raw and heated ruminant milk samples were presented, which were: αS1-casein, αS2-casein, β-casein, κ-casein, glycosylation dependent cell adhesion molecule 1 (GlyCAM1), and serum amyloid A (SAA or SAA3). The AUC of the non-redundant peptides in the skimmed raw and heated ruminant milk samples at time 0 h was used as a control.

#### 2.4.5. Potential Bioactive Peptide Search

The peptide sequence of the skimmed raw and pasteurised ruminant milk samples before and after self-digestion was used to search against the Milk Bioactive Peptide Database (MBPD) (http://mbpdb.nws.oregonstate.edu, accessed on 19 July 2022) [17] to identify potential bioactive peptides. The search type was sequence, and a similarity threshold of 100% was used to identify peptides with the same sequence to known functional milk peptides in the database.

The potential bioactive peptides were grouped by bioactivity for each combination of heat and self-digestion time within the species. The abundance of each bioactivity was calculated by the sum of the peptide AUC and divided by the total number of peptides identified in each functionality, which was named as “AUC/potential bioactive peptides”. The most abundant functionalities found in each species were reported, which were: antithrombin, antimicrobial, angiotensin converting enzyme (ACE) inhibitory, immunomodulatory, MUC4 expression, and calcium uptake. The AUC/potential bioactive peptides of each functionality in the skimmed raw and heated ruminant milk samples at time 0 h were used as a control.

### 2.5. Statistical Analysis

Statistical analyses were performed using the mixed model procedure of SAS (SAS/STAT version 9.4; SAS Institute Inc., Cary, NC, USA). A three-way ANOVA model that included the effect of milk treatments (raw, 63 °C/30 min and 85 °C/5 min), self-digestion time (0 and 6 h), species (bovine, ovine, and caprine), and all their interactions on free NH_2_ was used. The triple interaction was not significant and removed from the final model. The model diagnostics for each response variable were tested after combining the Output Delivery System Graphics procedure and the repeated statement of SAS, before comparing the means. The repeated statement in the mixed model procedure was used to test the homogeneity of variances by fitting models with the restricted maximum likelihood method and comparing them using the log-likelihood ratio test. Each response variable in the selected model had adjusted equal variances across treatments. Selected means were compared using the adjusted Tukey-test when the F-value of the analysis of variance was significant (*p* < 0.05). Principle component analysis (PCA) and partial least squares discriminant analysis (PLS-Da) on the protein abundance results were performed using the MixOmics (v6.14.1) package [18] in R (v4.0.2) [19]. Proteins were filtered based on a measured abundance in at least 75% of all samples within a species (bovine, caprine, and ovine) prior to PCA and PLS-Da.

## 3. Results

### 3.1. Degree of Protein Hydrolysis of Ruminant Milk Prior and Post Self-Digestion

There was a significant effect (*p* < 0.01) for the double interaction between species and heat treatments, species and digestion time, and heat treatment and digestion time (Table 1).

Among the skimmed raw ruminant milk, caprine had a 2.4- and 2.1-fold higher free NH_2_ concentration than raw bovine and ovine milk, respectively (Figure 1A). The heat treatments affected the free NH_2_ concentration in bovine and caprine milk. For instance, the free NH_2_ concentration decreased 59% in bovine milk heated at 85 °C/5 min and 44% and 53% in caprine milk heated at 63 °C/30 min and 85 °C/5 min, respectively, compared with their raw milk counterparts (Figure 1A).

After self-digestion, the free NH_2_ concentration increased 36%, independently of the heat treatment, for ovine milk only (Figure 1B) and increased 16 and 13% in skimmed raw and heated at 63 °C/30 min milk (Figure 1C), irrespective of the species.

### 3.2. Similarity between Non-Redundant Peptides Identified in Ruminant Milk Samples

The initial PCA (principal component analysis) of the non-redundant peptide sequences showed that two PC explained 52% of the variation, and it grouped the samples across species (bovine, ovine, and caprine) (Figure 2); despite that, the peptides of the heat treated samples prior and post self-digestion were included in each species.

Within species, the peptide sequences were analysed with PLS-DA (partial least-squares discriminant analysis) (Figure 3). In the PLS-DA score plots, there was clear separation of the clusters of skimmed raw ruminant milk prior and post self-digestion. However, this separation was only clear in ovine milk heat treated at 85 °C/5 min (Figure 3B).

### 3.3. Number of Non-Redundant Peptides in Heat-Treated Ruminant Milk

Prior to self-digestion (0 h), a total of 1146, 1735, and 1769 non-redundant peptides were identified in skimmed raw bovine (Figure 4A), ovine (Figure 4B), and caprine (Figure 4C) milk, respectively. Only 10, 12, and 22 of these peptides were found exclusively in skimmed raw bovine, ovine, and caprine milk, respectively, prior to self-digestion.

The combined data showed an increased total number of non-redundant peptides in the skimmed raw and heat-treated bovine and ovine milk after self-digestion. For instance, the number of non-redundant peptides in the raw bovine and ovine milk increased 7.4% and 7.7%, respectively, compared with prior to self-digestion. However, this increase was not observed in the raw and heated at 63 °C/30 min caprine milk. A total of 944, 1415, and 1248 common non-redundant peptides were identified between all heat treatments and self-digestion time combinations within the bovine, ovine, and caprine milk samples, respectively.

### 3.4. Protein of Origin

The breakdown of a few proteins was also tracked by studying the released peptides after the self-digestion of the skimmed raw and heat-treated ruminant milk samples (Figure 5).

After self-digestion, β-casein was the most hydrolysed protein in the skimmed raw and heated at 63 °C/30 min bovine mils, with 1.5 and 1.7-fold higher AUC, respectively. The same trend was observed in the raw ovine and caprine milk, but for the proteins αS1- and β-casein in ovine milk (1.9- and 1.2-fold) and β-casein, κ-casein, and SAA in caprine milk (3.7-, 3.2-, and 1.9-fold). In general, after self-digestion of the heated at 63 °C/30 min ovine and caprine milk, the proteins identified were not hydrolysed.

### 3.5. Potential Bioactive Peptides in Ruminant Milk after Self-Digestion

The identified peptides were compared to the MBPDB database to identify potential bioactive peptides in the skimmed raw and pasteurised ruminant milk samples prior to and post self-digestion. From the 1146 non-redundant peptides identified in raw bovine milk prior to self-digestion (Figure 4A), there were 59 peptides with potential bioactivity. Similarly, only a few non-redundant peptides (59 and 61) showed potential bioactivities in the raw ovine and caprine milk, respectively, prior to self-digestion. Post self-digestion, the abundance (AUC/potential bioactive peptides) of the most five abundant functionalities together had a 1.87, 4.56-, and 1.04-fold increase in the raw, 63 °C/30 min, and 85 °C/5 min bovine milk, respectively (Figure 6A). The same trend was observed in caprine milk. However, in ovine milk, the abundance only increased in the heated at 85 °C/5 min (1.22-fold) milk.

In the bovine and caprine milk, most of the potential bioactive peptides had antithrombin (37–56%, respectively), antimicrobial (25–14%, respectively), ACE-inhibitory (13–6%, respectively), immunomodulatory (15–20%, respectively), and calcium intake (9–4%, respectively) bioactivities. In ovine milk, “MUC4 expression” had a 2.5-fold higher abundance than “calcium intake”, making it one of the five top-ranked bioactivities and removing calcium intake from the list. Self-digestion increased 6.8-, 3.5-, and 5.4-fold and 3.3-, 2.7-, and 2.9-fold the antithrombin, antimicrobial, and immunomodulatory abundance (AUC/potential bioactive peptides) in bovine milk heated at 63 °C/30 min and raw caprine milk, respectively.

## 4. Discussion

In general, prior to self-digestion, free NH_2_ concentrations were more negatively impacted by the intense heat treatment (85 °C/5 min), which means that protein hydrolysis was promoted at this high temperature. However, post self-digestion, their concentrations increased (36%), independently of the heat treatment, for ovine milk. During self-digestion, β-casein, αS1-casein, κ-casein, and SAA were the proteins most hydrolysed. Moreover, the abundance of potential bioactive peptides increased mainly in bovine (heated at 63 °C/30 min) and caprine skimmed raw milk post self-digestion.

The highest concentration of free NH_2_ in the skimmed raw caprine milk before self-digestion was attributed to the highest number of non-nitrogen protein (i.e., non-redundant peptides), the concentration of proteases such as plasmin and cathepsin D [4], and different physicochemical properties and protein composition between the species. Caprine milk has a higher casein micelle size [2,20] and lower κ-casein concentration [21] than bovine and ovine milk. κ-Casein is recognised as providing stability to casein micelles and its lower concentration could have contributed to the increase in the exposure of micelles and other proteins (i.e., α-casein and β-casein) to the action of endogenous proteolytic enzymes during milking, partially explaining the highest concentration of free NH_2_ in skimmed raw caprine milk.

The decrease in the free NH_2_ concentration in the heat treated bovine (63 °C/30 min) and caprine (63 °C/30 min and 85 °C/5 min) milk might have occurred due to the aggregation of proteins and the Maillard reaction. Some proteins (e.g., β-lactoglobulin) contain disulphide bonds that can become reactive with free NH_2_ when the proteins are denatured during heating [5,22]. Moreover, Maillard reactions with amino groups and lactose occur during the heat treatments, which may have decreased the concentration of free NH_2_ in the milk samples [23,24]. Consequently, this reaction can potentially result in the body’s inability to efficiently utilise amino group–lactose complexes, leading to their excretion due to the body’s limited capacity to metabolise this compound effectively [25].

The reduction in free NH_2_ concentration due to heat treatment can also mean that milk proteins are undergoing decreased digestion by the native milk proteases [26]. This alteration can have a multifaceted impact influencing both the extent and speed of protein digestion, which can significantly influence the spectrum of bioactive peptides released during this process [27].

At post self-digestion, the proteins were shown to be more resistant to hydrolysis at 85 °C/5 min than in the skimmed raw and 63 °C/30 min milk. This finding is consistent with the results reported by Ren et al. [28], who observed that higher heat treatment (80 °C/30 min) resulted in a lower degree of protein hydrolysis during the gastric phase compared to lower heat treatment (65 °C/30 min). This effect could be explained by the destabilisation of the proteolytic system caused by the more intense heat treatment, where some active enzymes such as plasmin [3] could have been inactivated, reducing the differences between the free NH_2_ prior to and post self-digestion. Another possibility could be attributed to the formation of thermally induced aggregates between caseins or caseins and whey proteins [24,29], hampering the action of endogenous proteolytic enzymes [5,29].

The peptidomic data were used to better understand the protein hydrolysis by endogenous proteases within each milk type. The AUC of peptides during self-digestion was little affected in milk samples heated at 85 °C/5 min compared to skimmed raw milk and heated milk at 63 °C/30 min. This result supports the lower free NH_2_ concentration observed in milk heated at 85 °C/5 min prior to and post self-digestion. In the raw and heated at 63 °C/30 min samples, the proteins most hydrolysed during self-digestion were the caseins (α-, β- and κ-casein), suggesting that the increase in the free NH_2_ concentration during self-digestion could be due to the hydrolysis of casein.

According to our previous publications [3], plasmin, a protease known to especially hydrolyse β- and α-casein, was found in higher concentrations in the skimmed raw and heated at 63 °C/30 min milk compared to the milk heated at 85 °C/5 min. This finding suggests that plasmin may have played a significant role in the increased hydrolysis of β- and α-casein during the self-digestion of raw and heated at 63 °C/30 min milk. Moreover, caseins in raw and mild heat-treated milk are less resistant to hydrolysis due to the lower or no formation of thermally-induced aggregates between proteins and the lower number of Maillard reactions, facilitating the action of proteases [29].

A low AUC for peptides from whey proteins (i.e., β-lactoglobulin and α-lactalbumin) in all milk types suggest that these proteins are highly resistant to hydrolysis by endogenous enzymes. According to Guo et al. [30] and Chauvet et al. [31], the resistance of whey proteins to proteolysis is related to their conformation. Some whey proteins bind to components such as retinol (β-lactoglobulin), metal (lactoferrin), or lactose (α-lactalbumin) [30], which reduces the action of proteases.

Potential bioactive peptides were identified in skimmed raw and heat-treated ruminant milk prior to self-digestion. Self-digestion did not change the functionalities of the peptides such as antithrombin, antimicrobial, ACE-inhibitory, and immunomodulatory bioactivities. However, in general, self-digestion increased the abundance (AUC/potential bioactive peptides) compared to those in the control milk (time 0 h), showing that the endogenous proteases have an important role in protein digestion, resulting in differences in the hydrolysis and abundance of bioactivities between the heat treatments. For instance, the antithrombin abundance increased in bovine milk heated at 63 °C/30 min compared to prior and post self-digestion. The peptide LLYQEPVLGPVRGPFPIIV and multiple hydrolysis products (LYQEPVLGPVRGPFPIIV, YQEPVLGPVRGPFPIIV, and YQEPVLGPVRGPFPI) were probably caused by endogenous enzymes. Cathepsin D is a protease known to cleave amino acids such as leucine (L), isoleucine (I), and valine (V) located in the position P1 and P1’ [32], and so could be considered as one of the responsible proteases that hydrolysed the peptide LLYQEPVLGPVRGPFPIIV.

Interestingly, ovine milk heated at 63 °C/30 min had a reduced abundance of potential bioactive peptides despite an increase in the concentration of free NH_2_ after self-digestion, which indicated that greater hydrolysis does not always mean that more bioactive peptides will be released.

Some functionalities increased more than others during self-digestion. For example, in bovine milk heated at 63 °C/30 min, antithrombin and ACE-inhibitory bioactivities increased 7- and 2-fold, respectively, post self-digestion. The lower increase in the ACE-inhibitory compared to antithrombin bioactivity could be due to ACE-inhibitory peptides binding with proline (P) residues, which increased the peptide resistance to hydrolysis by proteases [17,33,34].

These bioactive peptides including antithrombin, antimicrobial, ACE-inhibitory, and immunomodulatory activities play crucial roles in human health, offering diverse physiological effects. For instance, antithrombin acts as an anticoagulant, reducing clot formation and thrombotic risks [35,36]. Antimicrobial peptides enhance innate immunity against bacterial pathogens [37,38,39], while ACE-inhibitory peptides regulate blood pressure [17,40,41]. Immunomodulatory peptides manage immune reactions and inflammation, bolstering overall resilience [17,42]. Integrating these bioactive peptides into dietary practices has demonstrated potential in improving human health [43,44,45].

In recent years, the production of bioactive milk ingredients has emerged as a dynamic sector for dairy and bio-industries [43,45]. Proteins, with their nutritional attributes and physiological functions driven by bioactive peptides, have garnered particular interest. Therefore, this study not only provides valuable insights into increasing the abundance of bioactive peptides with the application of self-digestion, but also holds potential for practical application in the industry. It is important to note that this study solely utilised a self-digestion model, which may not fully represent in vivo digestion. Consequently, when designing new products, the results of this study should be interpreted cautiously with regard to the increase and decrease in peptides abundance.

## 5. Conclusions

Prior to self-digestion, the heat treatments affected the free NH_2_ concentrations according to the species and the intensity of the treatments, which was reflected in the cleaved peptide bonds and peptide profiles released during self-digestion. Moreover, self-digestion, in general, increased the abundance of potential bioactive peptides in the skimmed raw milk and milk heated at 63 °C/30 min from ruminant species. This indicates that endogenous enzymes have an important role in the hydrolysis of milk proteins, although this role can be influenced by the heat treatment applied. The results of the study also suggest that heat treatment could be used to modulate self-digestion and the release of bioactive peptides.

To better understand the role of the proteases on the digestion of ruminant milk proteins, further studies are needed to determine the protease activities before and after the heat treatments. Future studies are also required to gain more insights into the effect of gastrointestinal conditions (i.e., pH and enzymes) on the activity of the milk proteases, which will reflect on the cleavage of peptide bonds and peptides released during gastrointestinal digestion.

## Figures and Tables

**Figure 1 foods-12-03511-f001:**
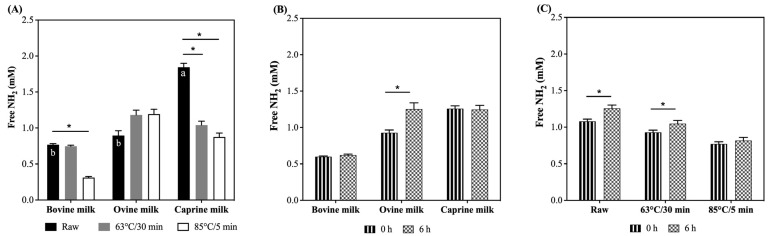
Free NH_2_ concentration in the raw and pasteurised (63 °C for 30 min and 85 °C for 5 min) bovine (**A**), ovine (**B**), and caprine (**C**) milk self-digested for six hours at 37 °C. Values are means ± SEM, n = 6. Raw and 0 h are the controls and refer to the milk before the heat treatments and self-digestion, respectively. There was a significant effect for the species (*p* < 0.001), heat (*p* < 0.001), and self-digestion time (*p* < 0.05) factors and the interaction between species and heat (*p* < 0.001) (**A**), species and self-digestion time (*p* < 0.05) (**B**), and heat and self-digestion time (*p* < 0.05) (**C**). Bars with an asterisk symbol express significant differences (*p* < 0.05) across the heat treatments within species (**A**), self-digestion time within species (**B**), and self-digestion time within heat treatments (**C**). Raw bars (**A**) with different letters indicate significant differences (*p* < 0.05) between the species.

**Figure 2 foods-12-03511-f002:**
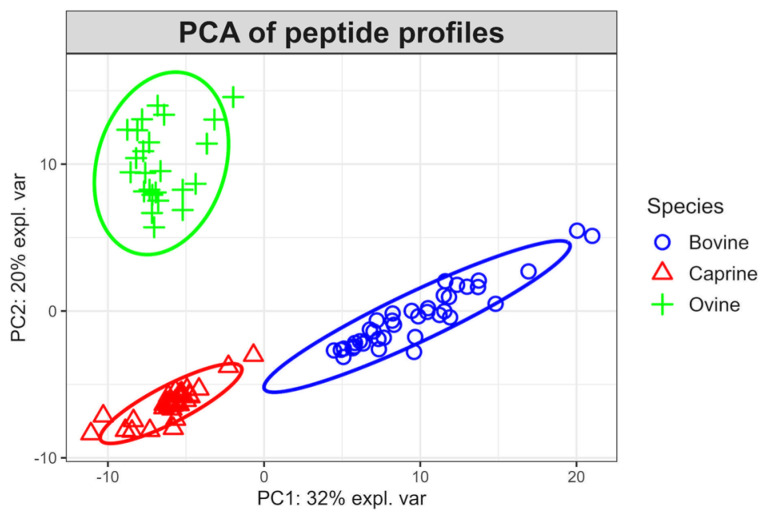
Principal component analysis (PCA) using non-redundant peptide sequences of the raw and heat-treated (63 °C/30 min and 85 °C/5 min) milk (bovine, ovine, and caprine) prior and post self-digestion.

**Figure 3 foods-12-03511-f003:**
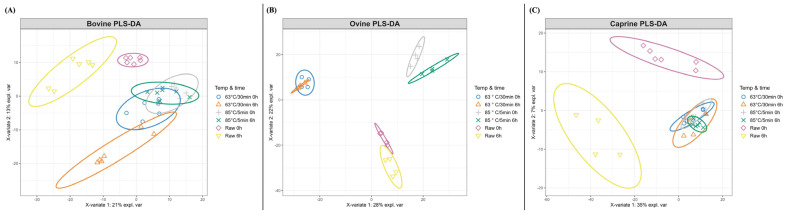
Partial least squares discriminant analysis (PLS-DA) using non-redundant peptide sequences of raw and heat treated (63 °C/30 min and 85 °C/5 min) milk (bovine (**A**), ovine (**B**), and caprine (**C**)) prior and post self-digestion.

**Figure 4 foods-12-03511-f004:**

Venn diagram showing the number of non-redundant peptide sequences that were unique and similar in the raw and heat-treated (63 °C/30 min and 85 °C/5 min) milk (bovine (**A**), ovine (**B**), and caprine (**C**)) prior and post self-digestion.

**Figure 5 foods-12-03511-f005:**
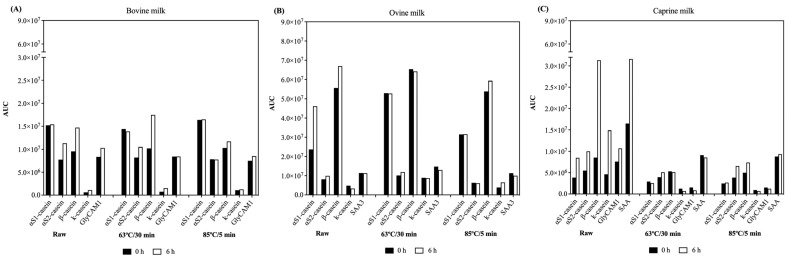
Area under the curve (AUC) of non-redundant peptides in the ruminant milk samples (bovine (**A**), ovine (**B**), and caprine (**C**)) prior (black bars) and post (white bars) self-digestion within each of the selected proteins.

**Figure 6 foods-12-03511-f006:**
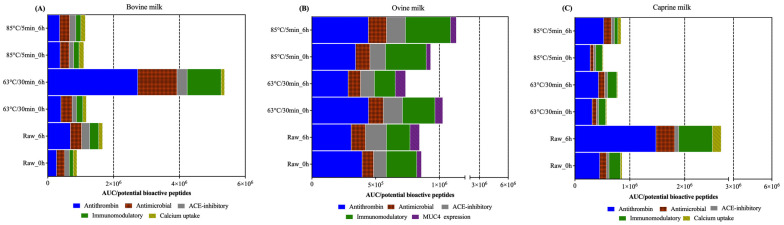
Area under the curve by the number of potential bioactive peptides (AUC/potential bioactive peptides) in the ruminant milk samples (bovine (**A**), ovine (**B**), and caprine (**C**)), which were enzymatically released from proteins prior and post self-digestion.

**Table 1 foods-12-03511-t001:** *p*-Values of factors remaining in the final model of an initial three-way ANOVA model to test the effect of milk species (bovine, ovine, or caprine), heating process (raw, 63 °C/30 min, or 85 °C/5 min), and self-digestion time (0 and 6 h) on the free NH_2_ concentration.

	Species (S)	Heating (H)	Self-Digestion (SD)	S × H	S × SD	H × SD	S × H × SD
Free NH_2_	<0.0001	<0.0001	0.01	<0.0001	0.01	0.01	- ^1^

^1^ The S × H × SD interaction was not significant (*p* > 0.05) and was removed from the final model.

## Data Availability

Data are available upon request from the corresponding author.

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
