# Peer review of "Effect of Heat Treatment on Protein Self-Digestion in Ruminants’ Milk"

_foods, 2023, doi:10.3390/foods12183511_

Round 1

Reviewer 1 Report

Comments to Authors:

1.       Provide the results of HR-LCMS in abstract. The abstract with quantitative will be encouraged.

2.       Line 48-52: change the position of word “heat-sensitive” in the sentence “Proteases are heat-sensitive, therefore mild treatments, such as 63°C for 30 min, can increase protease activities due to the activation of zymogens….”

3.       Line 63-64: provide logic for the selection of self-digestion temperature and time.

4.       Provide the merits and demerits of self-digestion and or heat treatment in introduction. How this study will help future studies or enhance understanding about the topic.

5.       Avoid un-necessary discussion or exaggeration of (line 62-73) in this study……..

6.       Line 76-78: provide GPS location for the samples collected.

7.       Specify the scientific name or variety of ovine and caprine from which milk obtained.

8.       Line 86: the holding time of heat treatment was recorded after achieving the specified milk temperature? Explain this clearly.

9.       Line 93: authors are talking about skimmed raw or just raw? Please specify clearly and change throughout the text if it was skimmed raw.

10.   Please double check too sure about “flow rate of 3,000 nL/min; flow rate of 600 nL/min”.

11.   Line 132-133: it will be nice to write just LCMS instead of “liquid chromatography mass spectrometry”.

12.   I appreciate authors for proper presentation of results and in flow discussion.

13.   It will be more impactful if authors revise the discussion part with latest citations.

14.   Give a paragraph on importance of milk peptides for human. How these peptides will be useful in healthy wellbeing?

Need to improve at some places.

Reviewer 2 Report

The research article "Self-digestion of proteins by endogenous proteases in raw and pasteurized ruminants’ milk" explores the impact of different heat treatments on the protein hydrolysis by endogenous enzymes in ruminants' milk (bovine, ovine, and caprine). The study uses a self-digestion model where milk is incubated at 37°C for six hours. The researchers measure free amino group (NH2) concentrations to assess peptide bond cleavage and identify peptide sequences using mass spectrometry.

Here are some points of improvement for the research article:

 Plagiarism Rate 37% (Must be reduced to permittable limit)

Title Improvement:

    • The title is quite verbose. It could be shortened and made more direct. An alternative could be: "Effect of Heat-Treatment on Protein Self-Digestion in Ruminants’ Milk".

Abstract:

    • The abstract is detailed but lacks a clear statement about the implications of the study or its broader context in the field.
    • It might benefit from a concise statement regarding the importance or relevance of the study findings for the dairy industry or nutritional science.

Keywords:

    • Consider adding more specific keywords, such as "bovine milk", "ovine milk", and "caprine milk", for better searchability.

Introduction:

    • The introduction provides background on the importance of milk proteins and bioactive peptides but lacks a clear statement of the research gap or objective. The hypothesis could be more explicitly stated, and the introduction should better highlight the gap in understanding regarding the role of endogenous milk enzymes on self-digestion.
    • While the introduction mentions the importance of milk as a nutrient source, it would be helpful to provide more context on the relevance of studying endogenous proteases and their role in protein digestion, especially from a health perspective.

Materials and Methods:

    • The methods section is detailed but could benefit from clearer organization and subheadings to make it easier to follow. Consider dividing the section into subsections such as "Milk Sampling," "Heat Treatment and Self-Digestion," "Peptide Analysis," and "Data Analysis."
    •  Clarify the selection criteria for the milk samples. Were they randomly chosen? Were there any specific characteristics or criteria that were considered?
    • Justify the choice of specific heat treatments. While 63°C for 30 min and 85°C for 5 min might be standard, it would be helpful for readers to understand why these were chosen.
    • The self-digestion model and the conditions chosen should be justified. Why was 6 hours chosen? Why was the temperature set to 37°C?
    • Ensure that any equipment or procedure that could have a commercial bias (e.g., specific brands of spectrometers) is justified in terms of its scientific relevance.
    • The methods section could benefit from more detailed descriptions of the tools, equipment, and software used. For example, specific settings or parameters for the high-resolution liquid chromatography-tandem mass spectrometry could be provided.
    • There is no clear mention of the control group in the experiment. If one was used, it should be indicated. If not, a justification should be provided.
    • The statistical analysis should include more details about the specific tests used, significance levels, and post hoc tests. Also, it's important to indicate whether the significance levels were adjusted for multiple comparisons.

Results:

    • Some of the percentages and statistics mentioned in the results could be better visualized using tables or infographics for clarity.
    • Ensure that all results are presented in a logical sequence, providing a clear narrative flow to the findings.

Discussion:

    • The discussion could be expanded to provide a more thorough interpretation of the results in the context of previous research. Discuss the implications of the findings on the understanding of milk protein digestion and potential health benefits. Also, address any limitations of the study and suggest directions for future research.
    • Dive deeper into the implications of the results. What do they mean for the dairy industry, nutritionists, or consumers?
    • Compare and contrast these findings with those from other studies in the field to give context and validate the results.
    • Address any limitations in the study and suggest directions for future research.

References:

    • Ensure all citations are consistent in format.
    • The referenced database, Milk Bioactive Peptide Database (MBPD), should be adequately described and its relevance to the study discussed.

Conclusions:

·  Conclude by summarizing the main findings and their implications.

·        Provide clear takeaways or recommendations based on the study's outcomes.

Ethical Considerations:

·        If relevant, mention any ethical considerations, such as institutional review board (IRB) approval or informed consent for human or animal subjects.

Language and Formatting:

    • Proofread the article for any grammatical or typographical errors.
    • Ensure consistent formatting in terms of headings, subheadings, and text.

Minor editing of English language required

Round 2

Reviewer 1 Report

Please accept